# A Systematic Review of Literature on Caregiving Preparation of Adult Children

**DOI:** 10.3390/ijerph20136295

**Published:** 2023-07-04

**Authors:** Chang Liu, Jing Hu, Xue Bai

**Affiliations:** Department of Applied Social Sciences, Faculty of Health and Social Sciences, The Hong Kong Polytechnic University, Hong Kong 999077, China; jingann.hu@polyu.edu.hk (J.H.);

**Keywords:** adult children, caregiving preparation, older parents, systematic review

## Abstract

With the increasing life expectancy and ageing population, long-term care has become an urgent policy issue worldwide. The informal care provided by family members, particularly adult children, is a key aspect of long-term care. However, socioeconomic transformations have resulted in changing family and demographic structures and increased geographic mobility, reducing the capacity of families to provide informal care and meet the caregiving needs of older adults. For ageing families, care preparation can be an effective method for coping with eldercare challenges, and care preparation is attracting increasing attention from researchers. This study seeks to conduct a systematic review for studies on caregiving preparation by adult children that were retrieved from six databases, to synthesise the available evidence, and to identify knowledge gaps and opportunities for future investigations. The characteristics and main themes of eighteen reviewed articles were identified and analysed. This study discussed various conceptualisations of caregiving preparation by adult children, the prevalence of caregiving preparation, the factors related to caregiving preparation, and the related consequences. On the basis of the systematic review findings, several limitations of the literature and directions for future research were proposed to promote care preparedness and the well-being of ageing families.

## 1. Introduction

Adult children are the most common source of assistance for older adults to achieve ageing-in-place [1]. Drawing on the data from the Health and Retirement Survey during waves of 1995 to 2010, it was estimated that approximately 17% of adult children in the United States will become caregivers at some point in their lives and will provide an average of 77 h of care per month [2]. According to the report from the National Alliance for Caregiving [3], 50% of surveyed caregivers were providing care for their parents or parents-in-law. In China, it is estimated that 90% of older people rely on their families for eldercare [4], with majority of this care being provided by their adult children.

Both anticipated and actual caregiving for parents can be sources of stress for adult-child caregivers. As the potential primary caregivers for their parents, adult children tend to be concerned about how much help they can provide in the future and whether they can manage the burden of providing care [5,6]. Studies also discovered that providing care to older parents can generate multidimensional stress and burden for adult-child caregivers, including financial, physical, and mental burden [2,3]. Since adult-child caregivers are more likely to be working [3], the conflict between their work and family roles is likely to contribute to their caregiving stress.

Studies investigated the care preparation activities that older adults engage in as a proactive coping strategy, and they have identified their effectiveness for mitigating future long-term care challenges [7,8,9]. Applying the theories of decision-making, problem-solving, and planning to everyday situations, Sörensen and Pinquart [10] distinguished four types of preparation activities and developed the Preparation for Future Care Needs Model for studying care preparation among older adults. The model encompasses the following care preparation domains: awareness/avoidance of future care needs, information gathering, decision-making, and concrete planning [11,12]. Individuals first anticipate or avoid the need for future care; thereafter, they begin to collect information, which involves actively seeking out information from other people or the media and making decisions about future care arrangements. The final step in their planning process is concrete planning, which refers to the activities that will help them implement their plans.

Scholars have also suggested that adult children should prepare themselves for their future caregiving role and participate in care preparation activities for effectively reducing care anxiety and preventing their future caregiving burden. While studies on adult children’s caregiving preparation have been conducted for decades, there is currently no systematic review on this topic. This study seeks to fill this gap by conducting a comprehensive overview of the literature in this field, synthesising the available evidence, and identifying knowledge gaps and opportunities for future investigations.

The following research questions will be addressed in the systematic review: (a) How have studies conceptualised the caregiving preparation of adult children since the emergence of this research topic? (b) To what extent do adult children engage in caregiving preparation? (c) What are the factors related to caregiving preparation and what are their possible effects?

## 2. Methods

### 2.1. Search Strategy

The EBSCOhost, PsycINFO, ProQuest, Web of Science, PubMed, and CNKI databases were systematically searched. A lexicon of keywords was generated after a pilot search of the selected databases. With the exception of CNKI, the following combinations of keywords were used to identify and retrieve studies on anticipatory care preparation from the databases: ‘adult child*’ AND (‘elder* care’ OR ‘filial care’ OR ‘caregiving’) AND (‘prepare* OR ‘plan*’). For the search of the CNKI database, the following combinations of keywords were used: ‘子女’ (adult child) AND ‘养老照顾’ (care preparation) AND (‘准备’ (preparation) OR ‘计划’ (plan)). Relevant studies dated from the inception of the databases until November 2022 were identified and retrieved. This broad period was applied because few systematic reviews have explored this research topic since its emergence.

### 2.2. Inclusion and Exclusion Criteria

Studies were included in the present review if they: (1) were reports, reviews, book chapters, conference papers, dissertations, or journal articles related to the caregiving preparation of adult children for their parents, (2) were published in English or Chinese, and (3) were published between the inception of the databases and November 2022. Studies were excluded if they were commentaries, unpublished reports, not empirical studies, or published in languages other than English or Chinese. Studies were also identified from the reference lists of the articles already included in the present review. The abstracts or summaries of all included studies were reviewed, and the eligibility of each study was determined by two researchers who evaluated the relevance of each study to the study topic.

### 2.3. Data Collection

Figure 1 presents the Preferred Reporting Items for Systematic Reviews and Meta-Analyses (PRISMA) diagram describing study selection. The initial literature search yielded 924 studies, of which 145 duplicates were removed. Subsequently, the abstracts of the remaining 779 studies were screened, and 726 irrelevant studies were excluded; thus, 53 articles were included in the full-text assessment. Next, a number of studies were excluded for various reasons; specifically, five studies did not have English or Chinese full text, six studies were not empirical studies, eighteen studies were related to actual caregiving experiences instead of care preparation, three studies pertained to the care preparation from the perspective of older adults, nine studies were related to end-of-life care planning, two studies were related to care transition, and three studies were about perceptions of care provision. Subsequently, 11 studies were retrieved from the reference lists and added to the present review. Consequently, a final sample of 18 articles was included in the present review. The screening and selection processes were conducted by two researchers, independently, and their results were cross-checked.

### 2.4. Data Analysis

First, two researchers independently evaluated the quality of the extracted studies by using the Mixed-Methods Appraisal Tool, which has been widely used in systematic reviews to evaluate the quality of qualitative, quantitative, and mixed-methods studies. Inter-rater reliability was determined by calculating intraclass correlation coefficients, and the results indicated a high inter-rater reliability (0.823). All the studies yielded an overall score of 3 or higher, indicating an acceptable research quality. Next, thematic analysis was conducted to synthesise the findings of the reviewed studies. All studies were read by two researchers. Data on the sources of the literature, themes, research methods, participants, and key findings of the reviewed studies were coded and summarised. Data synthesis was conducted by comparing the findings of the included studies and identifying differences and similarities.

## 3. Results

Table 1 lists the characteristics of the reviewed studies. Among the 18 reviewed studies, 83.3% (N = 15) were journal articles, and 16.7% (N = 3) were dissertations. Half of the studies (N = 9) adopted quantitative methods, including questionnaire surveys (N = 8) and experiments (N = 1), four studies employed qualitative methods, four applied mixed methods, and one study adopted a case study design. The systematic review findings were analysed based on three themes: (1) conceptualisation of caregiving preparation, (2) prevalence of caregiving preparation, and (3) the influencing factors for and consequences of caregiving preparation. The details on the participants, research methods, and the main findings of the reviewed studies are listed in Table 2. Table 3 provides a summary of the reviewed studies’ conceptualisations or measurements of the caregiving preparation conducted by adult children for their parents.

### 3.1. Conceptualisations of Care Preparation

Studies proposed various conceptualisations of the caregiving preparation of adult children. Of the 18 reviewed studies, 10 conceptualised it as a multiphase process. Bromley and Blieszner [13] identified four decision-making activities (i.e., consideration, discussion, planning, and decision-making) as the sequential steps of caregiving preparation. Hansson et al. [20] reported that adult children’s awareness regarding their parents’ needs progresses gradually over time; initially, they consider these needs, and then they learn about ageing and monitor their parents regarding specific areas of concern. Brown [14] described that caregiving preparation involves the dimensions of awareness, decision-making, and planning, whereas Fowler and Fisher [18] described that it comprises awareness, information collection, and discussion. Stolee et al. [28] conceptualised caregiving preparation as a process in which adult children converse with their parents regarding care planning, and this process also involves communication and decision-making.

Two studies specifically developed models for studying the caregiving preparation of adult children [25,30]. Sörensen [30] developed the Preparation for Caregiving framework to explain care preparation of multigenerational families by applying the life course perspective, role theory, and several theories of planning and decision-making derived from cognitive psychology. Caregiving preparation is defined as the mental and physical actions that are related to the four dimensions of anticipation, decision-making, concrete planning, and role socialisation, and they are performed by an individual before they assume the role of caregiver for an older individual [26].

On the basis of Sörensen’s [30] framework, the theory of proactive coping, and the literature on the traditional values of Mexican Americans and Hispanic/Latino cultures, Radina [25] investigated how Mexican Americans prepared for parent care and developed the Caregiver Preparation Model. Compared with Sörensen’s [30] Preparation for Caregiving Model, which focuses on both the caregiver and the care recipient, the Caregiver Preparation Model is centred on the care preparation of caregivers and comprises the two main components of decision-making and caregiver socialisation/preparation. Caregiver socialisation/preparation further consists of three secondary components, namely caregiver selection/designation, anticipation, and planning. As indicated by this model, the primary component of decision-making influences each of the three secondary components.

The rest of the studies mainly examined caregiving preparation as a single-step event (e.g., anticipation [15,16], consciousness [21], or discussion [17]), or as-expected care arrangements related to multiple care domains [19,22,23,29].

The measurements of caregiving preparation were mostly self-developed questions, and few studies have validated the scales used. Several studies used scales developed for measuring older adults’ care preparation [9,18,31] to assess adult children’s caregiving preparation.

### 3.2. Prevalence of Care Preparation

Based on the self-report of 103 adult daughters aged between 30 and 55 years, Conway-Turner and Karasik [15] discovered that almost all of them (99%) reported that they had considered the possibility of providing care to their ageing mothers. Approximately 68.3% of them reported that they had considered this issue frequently (sometimes almost daily), and this high level of consideration was unaffected by their age. Moreover, the mean age at which they started to consider this issue was 28 years, indicating the early occurrence of the anticipation process, which could be due to the high media coverage of caregiving responsibilities and the increasing visibility of the mother cohort caring for their own mothers [15]. In another study involving the same sample, Conway-Turner and Karasik [16] further reported that over 90% of the adult daughters anticipated providing advice or information if a crisis occurred. Sörensen and Zarit [27] conducted a study that examined a sample of 33 multigenerational families, and they reported that substantial numbers of both mothers and daughters in the sample anticipated the need for care for their grandmothers.

Bromley and Blieszner [13] surveyed 169 adult children aged between 20 and 59 years and discovered that over 80% of them had thought about the future dependency needs of their still-healthy parents and 39% of them had discussed the topic of future care with their parents. In a survey of an older sample, Hansson et al. [28] reported that 82% of 242 adult children (median age, 40 years) had started to seriously consider issues and concerns related to caring for their older parents. However, most adult children only engaged in the early stages of caregiving preparation (e.g., awareness, consideration, and anticipation), with few acquiring concrete planning experience related to making future caregiving arrangements [26,27]. Bromley and Blieszner [13] reported that ageing families rarely conducted preliminary planning and final decision-making related to caregiving preparation.

### 3.3. Influencing Factors and Consequences of Caregiving Preparation

The age and gender of adult children are associated with their caregiving anticipation [13,21]. Lieberman [21] investigated a sample of 807 adults who were aged between 20 and 70 years and had at least one living parent or parent-in-law. They reported that age, gender, and race were associated with the awareness of future caregiving; specifically, adult children who were older, female, or Black were more likely to acknowledge the dependency needs of their parents. They also discovered that those from working-class families were more likely than those from middle-class families to feel that discussing dependency issues of parents is appropriate.

For adult daughters, the order of birth, the characteristics of the sibling network, the geographical proximity between mothers and daughters, and the health status of the daughters have been reported to be significantly related to their caregiving anticipation [15,32]. Filial anxiety has been reported to be correlated with the likelihood of adult children discussing their parents’ preferences for future care and their coping efficacy [17]. Attitudes towards shared autonomy and ageing anxiety, concerns about the negative effects of caregiving, and perceptions regarding a limited future have been reported to be associated with caregiving preparation [18]. However, Bradley et al. [33] did not identify a significant correlation between adult children’s filial anxiety and the extent of their engagement in caregiving preparation.

In a small sample of 33 three-generation families, Sörensen [30] discovered that adult daughters’ anticipation of household help for their mother was influenced by adult daughters’ relationships with their own daughters, with affective solidarity moderating the association between filial responsibility and the anticipation of household help. In a separate investigation of the same sample, Sörensen [26] reported that whether adult daughters discussed future caregiving with other family members was influenced by their mother’s age, but not by health; additionally, the internal locus of control was revealed to be associated with increased preparation for caregiving. Furthermore, in a study of 141 adult children (mean age, 48.7 years) [9], securely attached individuals were marginally more likely than insecurely attached individuals to make plans. A comparison of individuals who are already providing care with those not currently providing care indicated that secure attachment has a greater influence on the preparation activities of individuals not currently providing care.

In a sample of 165 women who were prospective caregivers, Paulson and Bassett [24] discovered that filial obligation fully mediated the close attachment style and preparation for caregiving. Broomley and Blieszner [13] reported that the age, personal authority, and family stressors of adult children were positively correlated with whether they considered their parents’ future care needs. Specifically, daughters were more likely to consider care arrangements and engage in discussions with their parents, and family stressors and personal authority negatively and positively, respectively, influenced the likelihood of discussion. Miller [22] and Stolee et al. [28] asserted that information deficiency is the key barrier preventing adult children from conducting long-term care planning for their older parents.

Hansson et al. [20] reported that the awareness of ageing issues and caregiving anticipation among adult children were the most often triggered by the health crises of their parents and by their perceptions of vulnerability, associated with their parents’ psychological adjustment, personality, and support resources. The age of an adult child’s mother is also significantly correlated with their caregiving anticipation [15,32]. Bradley et al. [33] reported that the age of parents significantly influenced whether respondents engaged in minimal, partial, or substantial planning behaviours.

Few studies investigated the consequences of caregiving preparation. In a qualitative study, Stolee et al. [28] revealed that long-term care planning discussions can alleviate the anxiety of older adults regarding future care arrangements, helping them to establish a sense of control and access family support. Planning can also help family members gain knowledge on appropriate care arrangements, which can lead to increased calmness, help alleviate doubts, and eliminate stress and potential conflicts. Moreover, initiating conversations early on can reduce the barriers to effective decision-making. Sörensen and Zarit [27] reported that few respondents indicated that they engaged in care preparation activities; however, the respondents who were planners were more satisfied than nonplanners with the amount of discussion and planning with their families.

## 4. Discussion

The present study is the first to systematically review the literature on the caregiving preparation of adult children for their older parents, and synthesised research findings regarding the conceptualisation, prevalence, influencing factors, and potential consequences of caregiving preparation. The findings of the present study can provide future directions for this research topic and have implications for service providers and policymakers to design measures for supporting adult-child family caregivers.

The findings of the systematic review revealed that most studies conceptualised caregiving preparation as a multistage process. These studies’ multidimensional conceptualisations generally corresponded to the key components in the Preparation for Future Care Needs Model [11,12]. They mostly covered one or several components, including awareness (or consideration) of parents’ ageing or future care needs, gathering information related to eldercare, discussion or communication with older parents regarding future care, decision-making about parents’ future care arrangements, preliminary planning, and role socialisation.

It is noteworthy that majority of studies examining caregiving preparation explicitly or implicitly address the concepts of anticipatory role socialisation and proactive coping. These are two useful strategies for families to undergo transitions [34] and cope with potential stressors [31]. Derived from the role theory, role socialisation refers to the process of gradually learning the norms, scripts, attitudes, values, and subtle rules that an individual must acquire to effectively function in society [35]. Applied to the family context, anticipatory socialisation reflects the learning that adult children engage in before assuming the caregiver role and the application of the knowledge they acquired through learning. It also involves the acquisition of new abilities and skills; in some cases, it may involve changes to an individual’s reference or social group [34]. When a person can anticipate a situation and learn from their experience (or those of others), they are more likely to efficiently undergo transitions when they encounter similar situations in the future [34].

Moreover, according to proactive coping theory [31], people can anticipate or recognise potential stressors and act in advance to modify their effects [7]. Proactive coping involves five interrelated stages to detect and respond to potential stressors, namely, resource accumulation, attention to/recognition of potential stressors, initial appraisal, preliminary coping efforts, and elicitation and use of feedback [31,36]. The recognition of stressors related to future eldercare, such as parents’ ageing and anticipation of future care needs, may initiate the caregiving preparation. With the assumption that future caregiving tasks are stressful events, the proactive coping theory has been more frequently used to study the care preparation among older adults [7,12] than adult children.

Future studies may consider systematically examining adult children’s caregiving preparation experiences from the perspective of anticipatory socialisation and proactive coping. A more comprehensive and theory-guided conceptualisation of caregiving preparation activities can help future research better investigate adult children’s essential involvement in the different stages of the caregiving preparation process and examine the relationships among different components of caregiving preparation and their impacts. Understanding the intrinsic connections and transitions between the different components can help provide targeted interventions to promote the development and maturation of the whole process. Additionally, attention should also be paid to the interactions and communications between adult children and older parents at multiple stages of the care preparation process. Although some studies highlighted the communication and discussion between adult children and older parents, few adopted an intergenerational perspective to understand their different roles, mutual influences, and practical initiatives to care preparation. Examining adult children’s caregiving preparation experiences from an intergenerational perspective can provide a more comprehensive overview of care preparation that considers the familial context and how different generations influence each other. This would help to develop customised intervention programmes to assist these two groups in overcoming difficulties they encountered in the care preparation process.

The results of the systematic review also identified a lack of comprehensive and validated measurements for the entire care preparation process. Existing studies basically only assessed certain phases, especially the preliminary stage, such as the awareness of caregiving responsibility and how adult children consider providing care to their parents. The lacking evidence and assessment of subsequent stages, such as the final decision-making stage, leads to a lack of evaluation of the whole process. In view of the limitations in the validities of the measurements, future studies should consider developing and validating measures of comprehensive caregiving preparation activities of adult children.

Although the reported prevalence of caregiving preparation cannot be compared between studies considering the variations in the types of activities examined, most studies reported a high prevalence of caregiving preparation. Studies also highlighted that many adult children were anticipating future caregiving needs and responsibilities, but few adult children were making definite plans [27,33], as evidenced by the low frequency of preliminary planning and final decision-making [13]. This phenomenon could be due to the absence of apparent health problems and urgent care needs among the parents of these adult children. Further research is required to investigate the caregiving preparation experiences of adult children, to further explore the barriers and challenges to their caregiving preparations, which can serve as a basis for the development of key strategies for addressing contingencies.

The influencing factors for caregiving preparation have rarely been explored from a theoretical perspective, with only a few studies applying attachment theory or role theory to guide their investigations. Furthermore, considering that all the reviewed studies had a cross-sectional design, limited knowledge is available regarding how the caregiving preparation of adult children progresses over time and what are the causal effects of the aforementioned influencing factors. Future studies should adopt an integrative framework, preferably combined with a longitudinal research design, to investigate the predictors of multistep care preparation activities among adult children.

Notably, few studies examined the potential effects of the caregiving preparation of adult children. Although Western scholars have frequently discussed the potential benefits of care preparation in reducing the potential stress due to caregiving, there is limited empirical evidence supporting the role of care preparation in reducing the stress caused by anticipating or actually taking on the caregiving role among adult children. Thus, scholars should empirically investigate the potential effects of caregiving preparation, especially the reduction of caregiver stress, concerns about providing eldercare, and the enhancement of caregiving preparedness and other aspects of the well-being of adult children and their older parents.

Overall, the literature on the caregiving preparation of adult children is still limited and primarily focused on Western countries, and few studies have examined the care preparation activities and processes of Asian adult children. Considering that Asian societies emphasise filial piety and filial care, adult children play a crucial role in meeting the caregiving needs of older adults, and they are more likely than Western adult children to become primary caregivers of older parents. Cultural sensitivity is crucial to care preparation activities in Asia. Therefore, scholars should further investigate the unique characteristics of care preparation of Asian adult children.

## 5. Conclusions

The present study systematically reviewed the literature on the caregiving preparation of adult children for future eldercare needs. The identified studies have several limitations, including the lack of consistent conceptualisation of caregiving preparation, comprehensive examination of the process of caregiving preparation, intergenerational perspectives, validated measures, longitudinal research design, representative sampling, guiding frameworks for examining influencing factors and consequences, and the underrepresentation of Asian populations. Future studies should aim to address these research gaps to enhance the understanding of caregiving preparation and facilitate the development of interventions to promote care preparedness and well-being in ageing families. 

## Figures and Tables

**Figure 1 ijerph-20-06295-f001:**
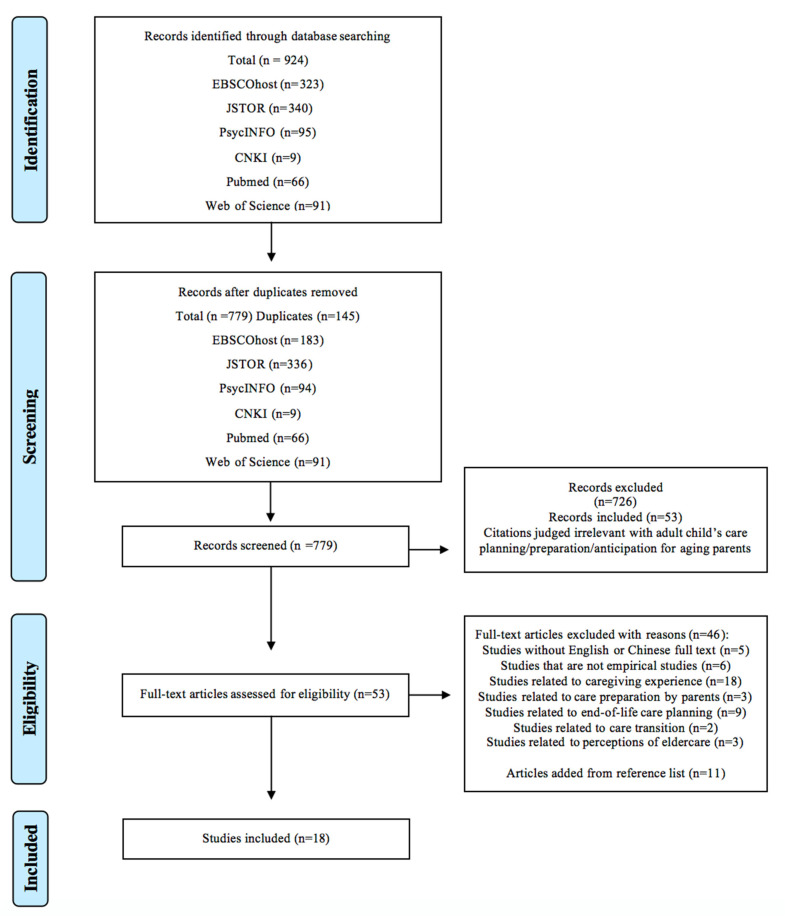
PRISMA flow of the literature search.

**Table 1 ijerph-20-06295-t001:** Characteristics of the reviewed literature on caregiving preparation of adult children for their parents.

Characteristics of Reviewed Literature (N = 18)	N	%
Source of literature		
Journal article	15	83.3%
2. Dissertation	3	16.7%
Method		
Quantitative: Questionnaire survey	8	45.5%
2. Quantitative: Experiment	1	4.5%
3. Qualitative: In-depth interview	4	22.7%
4. Mixed method: Questionnaire survey and interview	4	22.7%
5. Case study	1	4.5%

**Table 2 ijerph-20-06295-t002:** Descriptions of the reviewed literature on caregiving preparation of adult children.

Author (Year)	Participants	Method	Findings
Bromley and Blieszner (1997) [13]	169 adult children who were aged 20–59 and mostly White, highly educated, and female	Mail questionnaire survey	More than 80% of the adult children considered the future dependency needs of their still-healthy parent. The age, gender, personal authority, and family stressors of the adult children were associated with their caregiving anticipation. Preliminary planning and making final decisions rarely occurred. Family stressors and personal authority were related to the likelihood of conducting discussions.
Brown (2000) [14]	8 caregivers or caregiver-to-be, aged 40–60 years	Qualitative interviews and focus group	The process included three components, namely awareness, decisions, and thinking ahead, which influenced and were influenced by the feelings of participants.
Conway-Turner and Karasik (1993) [15]	103 adult daughters aged 30–55 years (mean age, 37 years)	Face-to-face questionnaire survey	In this study, 99% of the participants considered the possibility of providing care to their ageing mothers, 68.3% considered it either frequently or almost daily. The mean age at which they started to consider caregiving was 28 years. The proximity between mother and daughter, the health status of one’s daughter, and the age of the mother were associated with caregiving anticipation.
Conway-Turner and Karasik (1997) [16]	103 adult daughters aged 30–55 years (mean age, 37 years)	Face-to-face questionnaire survey	The daughters were more prone to anticipate providing care when a long-term crisis occurred than when a short-term crisis occurred. More than 90% of the daughters anticipated providing advice or information if a crisis occurred, whereas financial assistance or hiring others to help their mothers was anticipated more frequently in response to a long-term crisis.
Fowler and Afifi (2011) [17]	173 adult children aged >40 years (mean age, 47.44 years)	Experiment	Filial anxiety was correlated with the likelihood of engaging in discussions with parents regarding their preferences for future care and coping efficacy.
Fowler and Fisher (2009) [18]	128 adult children aged >40 years; 113 nonrelated parents aged >60 years	Face-to-face questionnaire survey	Attitudes toward shared autonomy and ageing anxiety, concerns about the negative effects of caregiving, and perceptions of a limited future were associated with caregiving preparation.
Gui and Koropeckyj-Cox (2016) [19]	20 Chinese young adults aged 23–31 years	Qualitative interviews	Some migrant adult children planned to settle down in Canada and bring their parents over, whereas others planned to return to China. The adult children were expected to take care of their parents in the future but also to consider their dilemmas.
Hansson et al. (1990) [20]	242 adult children (median age, 40 years)	Qualitative interviews and face-to-face questionnaire survey	In this study, 82% of the participants seriously considered issues and concerns related to caring for their older parents. A parent’s health crisis, perceptions of vulnerability associated with a parent’s psychological adjustment, personality, and support resources were associated with caregiving anticipation; 71% of the adult children intervened at some point, but such interventions were normally conducted in a supportive and conservative manner.
Lieberman (1978) [21]	807 adult children aged 20–70 years	Face-to-face questionnaire survey	The adult children’s age, gender, race, and social class were associated with their awareness of future caregiving.
Miller (2015) [22]	12 adult children aged 25–44 years	Case study	Information deficiency was the major barrier to the adult children’s long-term care planning for their older parents.
Myers et al. (2004) [23]	19 adult children (mean age, 53 years) from Alabama site; 11 adult children (mean age, 49 years) and 31 ageing parents (mean age, 74 years) from Texas site	Face-to-face questionnaire survey	Adult children and ageing parents rated several domains of parent care readiness, including the medical, legal–insurance–financial, family–social, and spiritual–emotional domains. The adult children’s emphasis on the domain of care preparation could differ from that of their ageing parents, but both generations still emphasised reconciliation within their families.
Paulson and Bassett (2016) [24]	165 women prospective caregivers aged 45–65 years	Online questionnaire survey	Filial obligation fully mediated the association between close attachment style and caregiving preparation.
Radina (2007) [25]	10 Mexican American sibling dyads	Qualitative interviews	The preparation process for parent care among Mexican Americans comprised the three conceptual components of caregiver selection/designation, anticipation, and planning.
Sörensen (1998) [26]	33 multigenerational families comprising grandmothers, mothers, and daughters	Qualitative interviews and face-to-face questionnaire survey	Few of the participants made concrete caregiving plans. The planners were more satisfied with the amount of discussion and planning that they engaged in with their family than the nonplanners were. The daughters’ discussion of future caregiving with their family members was predicted by their internal locus of control and the mother’s age.
Sörensen, Webster, and Roggman (2002) [9]	141 adult children (mean age, 48.7 years)	Face-to-face questionnaire survey	Attachment style was associated with caregiving preparation, which in turn was associated with feelings of preparedness.
Sörensen and Zarit (1996) [27]	33 multigenerational families comprising grandmothers, mothers, and daughters	Qualitative interviews and a face-to-face questionnaire survey	The daughters’ anticipation of household helping was predicted by their relationship with their own daughters, and affective solidarity moderated the association between filial responsibility and the daughters’ anticipation of household helping.
Stolee et al. (2014) [28]	24 older adults; 24 family members; 23 health and social service professionals; 3 representatives of key stakeholders	Qualitative interviews	The adult children believed that later-life care planning conversations alleviate the anxiety of older adults toward future care arrangements, thereby helping these older adults to establish a sense of control and access family support. Planning helped a family to learn about appropriate care arrangements. This awareness led to increased calmness among the families, helping them to alleviate their doubts and eliminate their stress and potential conflicts. The adult children required information to make informed choices but had limited access to information. Initiating conversations early could reduce barriers to effective decision-making.
Yoo and Kim (2010) [29]	124 adult children of immigrants aged 22–57 years	Qualitative interviews and a face-to-face questionnaire survey	The adult children felt a strong sense of responsibility toward their ageing parents, and they were prepared to support their parents’ financial, healthcare, and long-term care needs. Although both daughters and sons expressed the desire to care, daughters were more likely than sons to engage in in-depth discussions with their parents to clarify their concerns and worries.

**Table 3 ijerph-20-06295-t003:** Conceptualisations or measurements of caregiving preparation of adult children for older parents in the reviewed literature.

Author (Year)	Conceptualisations/Measurements
Bromley and Blieszner (1997) [13]	The decision-making process comprises four activities, namely considering, discussing, preliminary planning, and making a final decision. (1) Considering refers to whether an adult child has ever considered their parents’ future needs with respect to conditions that may limit the parents’ ability to live independently. (2) Discussing refers to whether an adult child has discussed their concerns with their parents. (3) Preliminary planning refers to activities involving obtaining information, asking for advice, or making arrangements for services if one’s parents cannot live independently. (4) Making a final decision refers to whether a plan is in place for the future care of one’s parents when necessary.
Brown (2000) [14]	Caregiving preparation comprises four main categories and several subcategories: (1) Child feelings refer to the feelings of an adult child regarding various decisions, emotional experiences, and expectations, and their feelings in regard to recognising change and loss. (2) Expectations refer to the expectations of an adult child’s parents and the adult child’s own expectations, roles, actions involving thinking ahead, feelings, and expectations. (3) Decisions refer to an adult child’s feelings regarding their decisions and their awareness of their parents’ feelings, emotional well-being, and physical well-being. (4) Recognition of change and loss refers to positive focus, family involvement, and comparisons.
Conway-Turner and Karasik (1993) [15]; Conway-Turner and Karasik (1997) [16]	In these studies, the planning and adjustment inventory was used. This inventory comprises information on when and how frequently the daughters investigated in these studies considered caring for their mothers (e.g., types of assistance, including financial assistance, emotional support, and help with daily living, frequency at which assistance is provided for short- and long-term illnesses, and likelihood of providing assistance).
Fowler and Afifi (2011) [17]	This study is based on the theory of motivated information management, which pertains to the decisions of individuals to seek out or avoid information regarding particular issues. The measurements used in this study include the importance of an issue (i.e., ‘knowing my parents’ future care preferences is essential’), the uncertainty discrepancy related to parental eldercare preferences (i.e., ‘how certain do you want to be about your parents’ preferences for their future care, and how certain are you about your parents’ preferences for their future care?’), outcome expectations (i.e., understanding the beliefs of respondents regarding the consequences of discussing future care with their parent that they identified at the beginning of the survey conducted in the study; the relevant items include ‘asking my parents what they think talking about this issue would lead to’ and ‘talking with my parents about what this issue would lead to’), and information-seeking behaviour (the level of information they had sought from their parents about care preferences (e.g., ‘how many questions have you posed to your parents regarding their preferences for care?’)).
Fowler and Fisher (2009) [18]	The four items from the ‘becoming aware’ subscale and four items from the ‘gathering information’ subscale of Sorensen and Pinquart’s (2001) measure of preparation for future care needs. Items of the original versions for older parents were reworded and applied to adult children (e.g., ‘the thought that my parent may need help or care in the future comes up a lot for me’). ‘Discussion of future care needs’ were measured by asking adult children to report on how much they had discussed with their parent a range of issues pertaining to future care needs.
Gui and Koropeckyj-Cox (2016) [19]	The expected care arrangements made by adult children for their parents.
Hansson et al. (1990) [20]	This study discussed the following concepts: (1) Participant-generated insights into processes (e.g., ‘if you have started to think about parent-caregiving issues, what was the event or moment that caused you to start to think along these lines?’ ‘what do you feel is the ‘right’ or ‘best’ time to begin sharing in or involving yourself in the decisions of your parents?’ and ‘what areas in your parents’ life are you monitoring more closely these days?’). (2) The likelihood of intervention index (i.e., the likelihood that a participant would intervene should problems arise across 6 domains (i.e., health, home maintenance, transportation, emotional support, finance, and bureaucratic mediation). (3) Elderly parent consciousness (i.e., a 12-item Likert scale that assesses the level of consciousness regarding ageing parents (e.g., the extent to which a participant had begun to think seriously about a parent’s changing needs, problems in activities of daily living, difficulties in living independently, etc.)). (4) The family involvement index (i.e., the extent to which a participant is now involved in providing care for their parents).
Lieberman (1978) [21]	This study discussed: (1) the perception of parent change (declining physical, psychological, and economic function), (2) the degree of parent concern (the extent to which a participant found parental changes to be challenging to cope with and sought help for them), and (3) the orientation of the participants toward behaviours related to parental ageing (measured using 25 items, including ‘asking your parents to confide in you more often when things are bothering them’ and ‘complaining to your brother or sister that you are taking too much of the burden of caring for your parents’).
Miller (2015) [22]	This study explored questions pertaining to the awareness of factors relating to long-term care planning and to risk aversion. It also examined how these concepts affected the way in which a participant advised their parents on the concept of long-term care planning (e.g., ‘how aware are you of the costs of long-term care in nursing homes, home healthcare settings, assisted care living communities, and adult day care?’ and ‘how confident are you in your financial ability to assist your parents in paying for long-term care if they need it?’).
Myers et al. (2005) [23]	A 50-item parent care readiness assessment instrument was used, covering the medical, legal–insurance–financial, family–social, and spiritual–emotional domains.
Paulson and Bassett (2016) [24]	Sörensen and Pinquart’s Preparation for Future Care Short Form (2002) was used (e.g., ‘I have gathered information about options for providing future help or care to my parent in the future’).
Radina (2007) [25]	The model employed in this study comprises two primary components: (1) decision-making and (2) the caregiver socialisation/preparation process.
Sörensen (1998) [26]	Guided by Sörensen’s [30] conceptual framework for caregiving preparation, this study examined the anticipation of caregiving, exploring whether a participant had anticipated: (a) their mother’s need for help with household tasks and (b) their mother’s need for help with personal care. It also examined (c) a family anticipation of providing care to the mother in the family.
Sörensen, Webster, and Roggman (2002) [9]	In this study, preparation for future caregiving is defined as the mental and physical actions involved in anticipating, making decisions, making concrete plans, and engaging in role socialisation relating to providing care for an older person in the future (Sörensen, [26]). Care planning was measured using the 14-item scale of the short form of Sörensen and Pinquart’s [11] Preparation for Future Care Needs Model.
Sörensen and Zarit (1996) [27]	Guided by Sörensen’s [30] conceptual framework of preparation for caregiving, the questions asked in this study were: (1) ‘Have you ever thought about what would happen if your mother needed help with household tasks like cooking or doing laundry or shopping?’ (2) ‘Have you ever thought about what would happen if your mother got sick and needed help with personal care like getting out of bed, taking a bath, or dressing herself?’ (3) ‘Have you/your family ever talked about what might happen if your mother needed help with household tasks or personal care?’ (4) ‘Have you and/or your family made any concrete plans in case your mother needs help with personal care or household tasks?’ (5) ‘How satisfied are you with the amount of discussion in your family about how to care for you if you ever needed it?’ (6) ‘How satisfied are you with the amount of planning in your family about how to care for you if you ever needed it?’
Stolee et al. (2014) [28]	In this study, three overarching themes were derived from interviews, namely role considerations, practical considerations, and emotional considerations. Under each of these themes, older adults, family members, and healthcare providers contributed different ideas.
Yoo and Kim (2010) [29]	In this study, the following key themes were derived through the application of grounded theory: (1) adult children were likely to talk about caring for their ageing parents in the future as a form of repayment for the care and support that they received from their parents while they were growing up, (2) adult children were concerned about the language and financial barriers faced by their ageing parents, and (3) adult daughters often felt more pressure to consider the details of their parents’ future caregiving needs.

## Data Availability

The data that support the findings of this study are available from the corresponding author upon reasonable request.

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
