# Peer review of "A Systematic Review of Literature on Caregiving Preparation of Adult Children"

_ijerph, 2023, doi:10.3390/ijerph20136295_

Round 1
Reviewer 1 Report (Previous Reviewer 1)
Thank you for the revised version of the manuscript titled, "A Systematic Review of Literature on Caregiving Preparation of Adult Children". The authors properly addressed my previous concerns.
Reviewer 2 Report (Previous Reviewer 2)
Dear authors,
thank you for your cooperation, I have no further comments. Wish you all the best in publishing.
This manuscript is a resubmission of an earlier submission. The following is a list of the peer review reports and author responses from that submission.
Round 1
Reviewer 1 Report
Major comments:
-
Did the authors consider following PRISMA guidelines when conducting this systematic review?
-
This systematic review did not evaluate included studies' quality. A quality appraisal is essential to a systematic review and should be added to the Methods/Results section. Also, the quality results should be presented in Table 2 or another table. Then based on the quality of the included studies, the authors should discuss current gaps and inform future research.
Minor comment
-
"Databases including EBSCOhost, PsycINFO, ProQuest, Web of Science, and CNKI were systematically searched." Pubmed was mentioned in Figure 1 but not here.
Reviewer 2 Report
Dear authors, thank you very much for the idea and effort in conducting this research.
However, I have few comments:
Abstract – please follow Aim, Methods, Results scheme so reader can easily go through it.
Methods: IPRISMA diagram – it is not clear eligibility section, as these number do not fit to tala number of studies, please make it more clear.
Why did you take so long period? In 45 years lots of concepts evolved, don’t you think this one also?
Also please specify implications for practice and how this can impact future research, do xou have some suggestions.
Although very interesting, I do not find straight focus in this research
English editing is necessary, together with grammatical errors correction.
Please follow journals guidelines for references